# Incidence of SARS-CoV-2 Infection and Related Mortality by Education Level during Three Phases of the 2020 Pandemic: A Population-Based Cohort Study in Rome

**DOI:** 10.3390/jcm11030877

**Published:** 2022-02-07

**Authors:** Laura Angelici, Chiara Sorge, Mirko Di Martino, Giovanna Cappai, Massimo Stafoggia, Nera Agabiti, Enrico Girardi, Simone Lanini, Emanuele Nicastri, Marina Davoli, Giulia Cesaroni

**Affiliations:** 1Department of Epidemiology of the Regional Health Service—Lazio, Via Cristoforo Colombo, 112, 00147 Rome, Italy; c.sorge@deplazio.it (C.S.); m.dimartino@deplazio.it (M.D.M.); g.cappai@deplazio.it (G.C.); m.stafoggia@deplazio.it (M.S.); n.agabiti@deplazio.it (N.A.); m.davoli@deplazio.it (M.D.); g.cesaroni@deplazio.it (G.C.); 2Clinical Division of Infectious Diseases, National Institute for Infectious Diseases Lazzaro Spallanzani-IRCCS, Via Portuense 292, 00149 Rome, Italy; enrico.girardi@inmi.it (E.G.); simone.lanini@inmi.it (S.L.); emanuele.nicastri@inmi.it (E.N.)

**Keywords:** SARS-CoV-2 infection, incidence, mortality, socioeconomic factors, inequality, education, epidemiology

## Abstract

Evidence on social determinants of health on the risk of SARS-CoV-2 infection and adverse outcomes is still limited. Therefore, this work investigates educational disparities in the incidence of infection and mortality within 30 days of the onset of infection during 2020 in Rome, with particular attention to changes in socioeconomic inequalities over time. A cohort of 1,538,231 residents in Rome on 1 January 2020, aged 35+, followed from 1 March to 31 December 2020, were considered. Cumulative incidence and mortality rates by education were estimated. Multivariable log-binomial and Cox regression models were used to investigate educational disparities in the incidence of SARS-CoV-2 infection and mortality during the entire study period and in three phases of the pandemic. During 2020, there were 47,736 incident cases and 2281 deaths. The association between education and the incidence of infection changed over time. Till May 2020, low- and medium-educated individuals had a lower risk of infection than that of the highly educated. However, there was no evidence of an association between education and the incidence of SARS-CoV-2 infection during the summer. Lastly, low-educated adults had a 25% higher risk of infection from September to December than that of the highly educated. Similarly, there was substantial evidence of educational inequalities in mortality within 30 days of the onset of infection in the last term of 2020. In Rome, social inequalities in COVID-19 appeared in the last term of 2020, and they strengthen the need for monitoring inequalities emerging from this pandemic.

## 1. Introduction

The novel coronavirus severe acute respiratory syndrome (SARS-CoV-2) causing coronavirus disease 2019 (COVID-19) appeared in late 2019 in Wuhan, China and rapidly spread progressively on a global scale, with the WHO declaring it a public health emergency of international concern [1]. On the basis of the published literature, men, people of advanced age, and those with underlying illnesses such as diabetes mellitus, hypertension, and cardiovascular disease tend to be susceptible to COVID-19 and at higher risk for adverse outcomes from COVID-19 [2].

Like other infectious diseases, the SARS-CoV-2 infection is likely to hit disadvantaged people harder [3,4]. It has been hypothesized that the COVID-19 pandemic can exacerbate inequalities in our societies through several direct or indirect pathways [4]. The most socially disadvantaged people potentially present a higher risk of infection. Usually, they have limited ability to practice effective physical distancing, have poor housing conditions, and they are at higher risk of worse complications and outcomes because of underlying social, age-related, and clinical vulnerabilities [4]. In addition, they may suffer more from the indirect effects of the pandemic as the reorganization of the health care system and the consequent reduction of planned services, including those involving chronic conditions whose burden is socially patterned [5]. Moreover, another possible indirect pathway concerns the effects of COVID-19 emergency lockdowns. The medium- and long-term socioeconomic consequences of the partial suspension of productive and economic activities, such as rising unemployment and poverty rates, are likely to affect more those in already poor conditions and amplify social and health inequalities [6]. The recent literature has begun to investigate the influence of social determinants on the evolution and impact of the COVID-19 pandemic [7,8]. Socioeconomic inequalities in the incidence of SARS-CoV-2 infection were reported in the United States [9,10], Spain [11,12], and Chile [13]. Several ecological studies showed that the number of cases and of COVID-19-related deaths were high in counties or regions characterized by socioeconomic deprivation or income inequality [14,15,16]. A number of studies investigated socioeconomic inequalities in COVID-19 mortality, showing a high burden in the disadvantaged groups of the population [5,17]. However, monitoring the effect of social determinants of health on the risk of becoming infected and experiencing adverse outcomes is crucial and has enormous repercussions on the effectiveness of policies implemented to control the epidemic [7,18]. The level of education is recognized as one of the critical components among these determinants, and its role needs to be explored [19].

The association between socioeconomic inequalities in the COVID-19 pandemic may be changing over time, presenting multiple stages [9,20]. In the early stages of the pandemic, the insufficient knowledge to create a defense can result in unclear or even unexpected social gradients in the infection and disease outcomes. When the availability of scientific evidence increases, health inequalities tend to grow, as advantaged groups of the population can react to control risks and adhere to guidelines more than the disadvantaged can [20]. Lastly, as health knowledge becomes widely available, together with preventive measures and treatments, socioeconomic inequalities decrease and eventually disappear [20]. Vaccination against COVID-19 was not available during 2020. Hence, we hypothesized that educational inequalities grew during the 2020 COVID-19 pandemic in Rome, and regarded both the incidence of infection and mortality among infected cases.

This work investigates the effect of disparities in educational level on the incidence of SARS-CoV-2 infection and mortality within 30 days of the onset of SARS-CoV-2 infection in the city of Rome in three different phases of the epidemic, reflecting differences in the management capacity of the health system and public awareness of the danger and spread of the virus.

## 2. Materials and Methods

### 2.1. Data Sources

The cohort of all residents in Rome as of 1 January 2020 was identified. The cohort, created to investigate COVID-19 determinants and consequences within the DeteCOVID project (funded by the Italian Ministry of Health), was based on the Lazio Region Longitudinal Study and Integrated Surveillance System of SARS-CoV-2 infections record linkage. The Lazio Region Longitudinal Study is the administrative cohort of all residents, obtained by the record linkage of census data with the Regional Health Information System, which comprises all health administrative databases, including patients, hospital discharges, and mortality registries. The Lazio Region Longitudinal Study is part of the National Statistical Program and was approved by the Italian Data Protection Authority. The Integrated Surveillance System of SARS-CoV-2 infections, established in the Lazio region at the beginning of the pandemic, collects individual data on all notified positive tests for SARS-CoV-2, with the date of incidence of infection and recovery. The linkage between the Lazio Region Longitudinal Study and the Integrated Surveillance System of SARS-CoV-2 infections was performed within the DeteCOVID project and acknowledged by the INMI Spallanzani Ethics Committee (29 July 2020). The linkage produced a dataset with anonymized identifiers, stored and processed on Lazio Region servers under strict controls to protect personal data.

### 2.2. Study Design and Population

A population-based prospective cohort study was performed. The cohort of participants resided in Rome on 1 January 2020 and were 35 or older, and available information on attained education was considered. Residents aged 35 or older were selected because information on educational level was obtained from the 2011 census, and education can be considered to be stable after the midtwenties [19].

### 2.3. Outcomes and Follow-Up

Outcomes of the study were incidence of SARS-CoV-2 infection and mortality within 30 days of infection onset (date of the first positive swab). Therefore, all residents were followed from 1 March to 31 December 2020, at date of infection or death (whichever came first). In addition, all cases for 1 March–31 December 2020 were followed for 30 days from the onset of infection.

Three different phases of the pandemic were considered: March–May, June–August, and September–December 2020. The first phase was characterized by limited treatments, testing, and knowledge of risk factors. The second was the summer, characterized by warm temperatures and limited infection spreading. Lastly, the third phase was characterized by increased cases and improved knowledge about the virus, transmission modes, and treatments. In each phase, patients not infected at the beginning of the period were followed until the date of infection, death, and end of the period (whichever came first). Similarly, patients with a date of first SARS-CoV-2 infection in each phase were followed from the date of infection for 30 days.

### 2.4. Exposure and Covariates

Exposure was educational level classified as low (<upper secondary school), medium (upper secondary school), and high (academic degree). As covariates, sex and age classes were considered (35–44, 45–54, 55–64, 65–74, 75–79, 80+).

### 2.5. Statistical Analysis

Descriptive analysis according to educational qualification, sex, and age classes was performed. The numbers of SARS-CoV-2 infections and deaths within 30 days of infection onset, crude cumulative incidence, and crude mortality rate with 95% confidence intervals (95% CI) were presented. Crude mortality rates were defined as the proportion of people at risk who had died of COVID-19.

Univariate log-binomial regression models were used to estimate the crude cumulative incidence of SARS-CoV-2 infection and mortality rate within 30 days along with 95% CI by educational qualification, sex, and age classes in the entire period of the study and the three phases of the pandemic. The chi-squared test was applied to compare the crude cumulative incidence and crude mortality rate in the different categories of the considered variables. Bonferroni adjustment was used to take into account multiple comparisons.

Age- and sex-adjusted log-binomial regression models were used to estimate the adjusted cumulative incidence of SARS-CoV-2 infection and mortality rate within 30 days, along with 95% CI in the study period and the three phases of the pandemic.

To investigate the association between educational qualification and incidence of SARS-CoV-2 infection, univariate and age- and sex-adjusted log-binomial regression models were used by estimating relative risks (RR, 95% CI, *p* value) by educational level in the entire period and the three different phases.

Lastly, Cox regression models adjusted for age and sex were used to investigate the association between educational attainment and mortality within 30 days of infection by estimating hazard ratios (HR, 95% CI, *p* value) by educational level for the entire period and each of the three phases of the pandemic. *p* values were adjusted by the mean of Bonferroni adjustment to account for multiple comparisons. Statistical significance was set at a 2-tailed *p* value of 0.05. All data were analyzed using SAS version 9.4 (SAS Institute, Cary, NC, USA).

## 3. Results

### 3.1. Incidence and Mortality Rate by Educational Level, Sex, and Age

The study population included 1,538,231 individuals (55.5% women) with a mean age of 60.2 ± 14.7 years. The number of participants excluded from the study because of missing data on educational level was 251,992. They did not differ from the study cohort in terms of sex composition (51% women), but were found to be slightly younger, with a mean age of 51.3 ± 13.3 years.

Table 1 shows that 47,736 incident cases with a mean age of 58.4 ± 14.7 years were identified over the entire study period. Out of the total of incident cases, 41.9% had a low level of education. The crude cumulative incidence in the entire period was 31.0‰ and was higher for low and medium compared to high educational qualifications (31.7‰ vs. 28.5‰, *p* value < 0.001 and 31.9‰ vs. 28.5‰, *p* value < 0.001) in males compared to females (32.7‰ vs. 29.7‰, *p* value < 0.001), and was higher for the youngest compared to the oldest age group (34.9‰ vs. 28.9‰, *p* value < 0.001) (Table 1 and Appendix A).

In the entire considered period, cases of SARS-CoV-2 infection resulted in 2281 deaths within 30 days of the start of infection with a mean age of 79.8 ± 10.5 years. Of the total number of infected cases who died, 70.8% had a low level of education. Crude mortality rate was 47.8‰ in the entire period, and was higher for low compared to high educational level (80.7‰ vs. 22.8‰, *p* value < 0.001) in males compared to females (57.4‰ vs. 39.3, *p* values < 0.001), and in the oldest compared to the youngest subjects (236.0‰ vs. 1.2‰, *p* value < 0.001) (Table 2 and Appendix A).

Figure 1 shows that the crude cumulative incidence (Figure 1A) was higher from September to December compared to the other two periods of March–May and June–August. Overall, the crude cumulative incidence of infection was higher in the low- and medium-educated compared to the highly educated, but in the first phase from March to May, a higher cumulative incidence was observed for high educational qualifications and older age groups. This also shows (Figure 1B) that the mortality rate was higher in the first period of March–May: 157.0‰ (95% CI 141.2–174.5) compared to the other two: 41.2‰ (95% CI 29.6–57.4) in June–August, and 37.7% (95% CI 35.9–39.6) in September–December. Results overlapping with those of the whole period were observed for the mortality rate by educational level, sex, and age classes in the three phases, except in June–August, when females had a higher mortality rate than that of males.

Results from age- and sex-adjusted analysis showed an adjusted cumulative incidence of 30.8‰ (95% CI 30.5–31.1‰) in the whole period with confirmed higher values in the third period of 28.3‰ (95% CI 28.0–28.5‰). The adjusted mortality rate was 17.6‰ (95% CI 16.2–19.0‰) in the whole period with confirmed higher values in the first period 87.4‰ (95% CI 73.7–103.5).

### 3.2. Association between Educational Level and SARS-CoV-2 Infection

Over the whole considered period, having a low or medium level of education compared with a high level was found to be significantly associated with an increased risk of infection, with RR = 1.23 (*p* value < 0.001) and RR = 1.12 (*p* value < 0.001), respectively. In the three phases, a reduction in the risk of infection for the low or medium level of education was found in the first phase: RR = 0.71 (*p* value < 0.001) and RR = 0.64 (*p* value < 0.001), respectively; nonsignificant results were found in the second phase. In the third phase, an increased risk was confirmed for the low or medium level of education compared to the high level: RR = 1.27 (*p* value < 0.001) and RR = 1.15 (*p* value < 0.001), respectively (Figure 2 and Appendix A).

### 3.3. Association between Educational Level and Mortality within 30 Days

In the entire considered period, having a low level of education compared with a high level was found to be significantly associated with an increased risk of mortality within 30 days of infection, with HR = 1.37 (*p* value < 0.001). Although the hazard ratio for medium vs. high level of education was greater than 1, it was not statistically significant (HR = 1.14; *p* value 0.1197). In the three phases, significant results were only found in the third phase with an increased confirmed risk for the low level of education compared to the high: HR = 1.43 (*p* value < 0.001) (Figure 3 and Appendix A).

## 4. Discussion

In Rome, the 2020 COVID-19 pandemic was characterized by two waves before and after the summer period. The second wave had a much higher cumulative incidence of infection than that of the first wave. The adjusted mortality rate was higher in the first wave than that in the second. Overall, there was strong evidence of socioeconomic inequalities in the incidence of SARS-CoV-2 infection, but the association between level of education and the incidence of infection changed over time. During the first phase, from March to May, low- and medium-educated individuals had a lower risk of infection compared to highly educated residents. During the second phase, from June to August, there was no evidence of an association between attained education and incidence of infection. Lastly, from September to December, there was evidence of socioeconomic inequalities in both the incidence of infection and mortality within 30 days from infection.

The SARS-CoV-2 epidemic curve in Rome was similar to the Italian epidemic curve, with a modest wave from February to April, which predominantly hit the northern area of the country, and a much more serious wave from September to the end of the year [21]. The cumulative incidence of infection was slightly higher in men than that in women. Men were also more likely to die within 30 days from infection than women were. A major vulnerability of men to COVID-19 was reported, with higher hospitalizations, ICU admissions, and mortality of men than those of women [22]. While the cumulative incidence was higher in the young age groups, mortality within 30 days from infection increased with age. The association between age and adverse outcomes of COVID-19 was extensively reported [2]. In 2020, the COVID-19 epidemic in Rome hit disadvantaged groups of the population harder. The results of this study showed socioeconomic inequalities in both incidence and mortality within 30 days from infection, confirming international results [10,13,14,23].

Although in the early days of the epidemic, the media hypothesized that the pandemic would become a great leveler, hitting the population without distinctions, it soon became clear that it would be the opposite, as in the historical epidemics of the past, from the 1630 plague to the 1930 Spanish influenza or the 2009 H1N1 influenza pandemic [4,18]. Socioeconomic position influences the neighborhood where a person lives and the type of housing, the number of cohabitants, and type of job. Only white-collar workers had the possibility of working remotely, and the majority of essential service workers (for example, those with jobs in food, delivery, and transportation) are not highly educated. Moreover, disadvantaged groups of the population have a high prevalence of chronic diseases such as hypertension, diabetes, obesity, COPD, cardiovascular diseases, all factors associated with adverse outcomes of COVID-19 [4,24]. Regarding the interplay of noncommunicable diseases and COVID-19, Bambra and colleagues argued that we are experiencing a syndemic pandemic, where risk factors and comorbidities in the most disadvantaged interact to exacerbate the disease and to worsen social conditions [4].

In this study, attained education was used as an indicator of socioeconomic position. Education can impact occupation, income, and economic stability, and represents intellectual resources. Given the inter-relationship among social determinants of health, it is difficult to disentangle the role of one compared to the others, and low socioeconomic groups of the population differ from high socioeconomic groups not merely on the investigated dimension. However, education as an indicator has its own peculiarity. It can be considered to be stable in adulthood [19] and is strongly associated with several health outcomes [25]. The skills attained in the educational process can affect a person’s cultural literacy, make them more receptive to health education messages, communicate with healthcare providers, and access appropriate health services [19]. Moreover, cultural literacy is strongly related to health literacy, defined by Galobardes and colleagues as “the degree to which individuals have the capacity to obtain, process, and understand basic health information needed to make appropriate health decisions” [19]. Health literacy is the ability to read and understand health-related pamphlets, written instructions from healthcare providers, prescriptions, and to understand the characteristics of infectious diseases, including the mode of transmission, the individual responsibility of adhering to social distancing, and other recommended measures, such as wearing masks or washing hands, and the reasoning behind the measures being taken to prevent the spread of the virus [26]. Education as a dimension of socioeconomic position has some limitations. As the literacy of a population increases with generations, i.e., younger generations are more educated than the older ones, the meaning of education changes across birth cohorts. For this reason, models were adjusted for age.

The socioeconomic pattern in SARS-CoV-2 infection and COVID-19-related mortality changed over time. Regarding the incidence of infection, in the first phase of the epidemic, the highly educated had a higher risk than that of low- and medium-educated residents. During the second phase, there was no evidence of an association between individual level of education and incidence of infection, whereas in the third phase, there was strong evidence of an inverse association. The different pattern in time was reported in Spain and US [9,12]. Clouston and colleagues provided a conceptualization about the rise and fall of social inequalities in disease over time, fundamental cause theory, and its extensions [9,20]. According to this theory, the first stage is characterized by a lack of knowledge about risk factors, possible preventive measures, or treatments. In this phase, the association between socioeconomic position and occurrence of the disease might be either a direct or an indirect association. The second stage is characterized by an unequal diffusion of innovative treatments, and inequalities are produced; the third stage is characterized by increased access to health knowledge, thus reducing inequalities. Lastly, a stage is characterized by widely available prevention and effective treatments that reduce mortality and eliminate the disease. Applying this conceptualization to the pandemic in Rome, the first phase was characterized by a lack of preparedness of both the population and healthcare providers. It was not surprising that the highly educated had a high risk of infection because, at the beginning, the diffusion of the virus circulating in hospital settings placed doctors and nurses (the highly educated) at the highest risk. Then, the lockdown from 11 March reduced the possibility of infections. Till the end of March, testing for SARS-CoV-2 infection was not widely available; hence, cases during the first phase could have been underestimated. In the second phase, during the summer, the infection circulated most among youth residents, with a mean age of cases of 57.5 ± 14.9 years versus 66.3 ± 16.0 years in the first phase. The third phase could be assimilated to the second stage of Clouston’s conceptualization, in which vaccination had not yet been offered, but knowledge on transmission mode was available, as was the differential possibility of social distancing. Regarding the association between level of education and 30 days mortality in infected patients over time, there was only evidence of an inverse association in the last period of 2020, confirming that, in September, we entered the fundamental cause theory’s second stage [9,20].

This work has its strengths and limitations. The first strength was the huge study population, which was composed of the entire population of Rome aged 35 years or older, with available information from the 2011 census linked to the Coronavirus Emergency Registry and Health Information System. The second was the use of the level of education as an individual index of socioeconomic position. Although educational level in adulthood is the strongest individual socioeconomic position indicator [19], it is rarely available in administrative data, and only a few settings as European Nordic countries could utilize it [27]. The use of attained education is also a limit of this study because the information was only available for those who had responded to the 2011 census of population. Hence, the study population represents the long-term residents of the city.

## 5. Conclusions

Our results confirm the fundamental cause theory expressed by Clouston et al. [9,20], showing the existence of social inequalities in COVID-19 incidence and mortality, which appeared in the third term of the 2020 pandemic in Rome. These results strengthen the need for monitoring inequalities emerging from this pandemic and for targeting disadvantaged communities in interventions aimed at preventing and alleviating the pandemic consequences. It is remarkable that the higher impact of COVID-19 in the most vulnerable individuals increases their health-related disadvantages [18]. Therefore, there is the urgency of policies for reducing the gap in health and access to effective healthcare treatments. Strategies to mitigate the impact of new pandemic outbreaks or similar disasters, particularly focused on low socioeconomic groups, might include increasing availability and accessibility of testing, contact tracing, isolation options, preventive care, disease management, and prevention guidance. The dramatic experience of the COVID-19 pandemic reinforces the need for social and economic policies designed to reduce the risk of disease, and promote universal and equal access to healthcare.

## Figures and Tables

**Figure 1 jcm-11-00877-f001:**
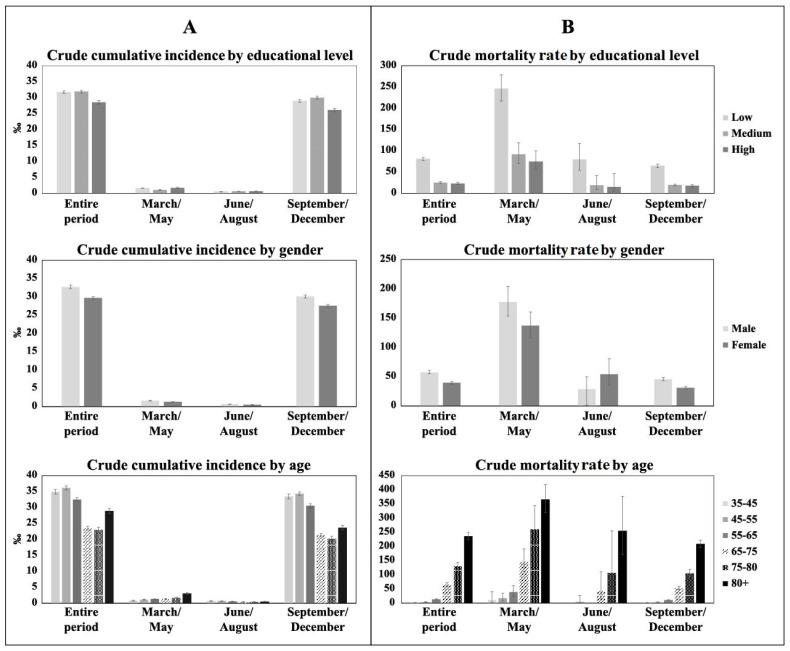
(**A**) Crude cumulative incidence and (**B**) crude mortality rate by educational level, sex, and age for the entire period and in the three phases of the pandemic.

**Figure 2 jcm-11-00877-f002:**
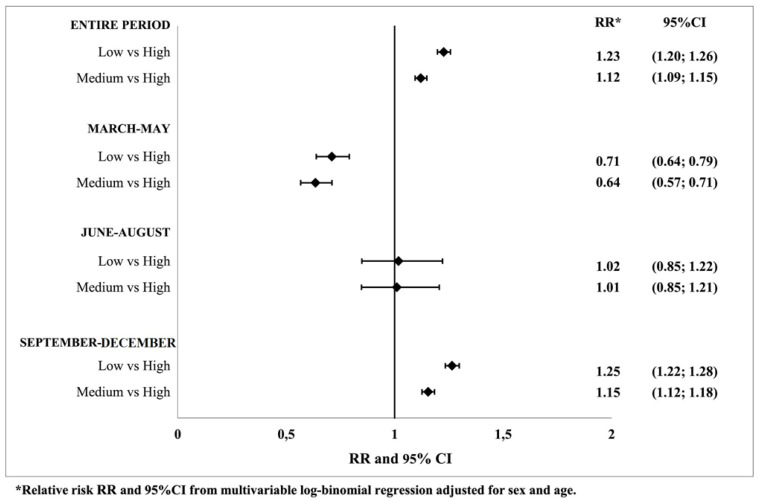
Association between educational level and SARS-CoV-2 infection for the entire period and in the three phases of the pandemic.

**Figure 3 jcm-11-00877-f003:**
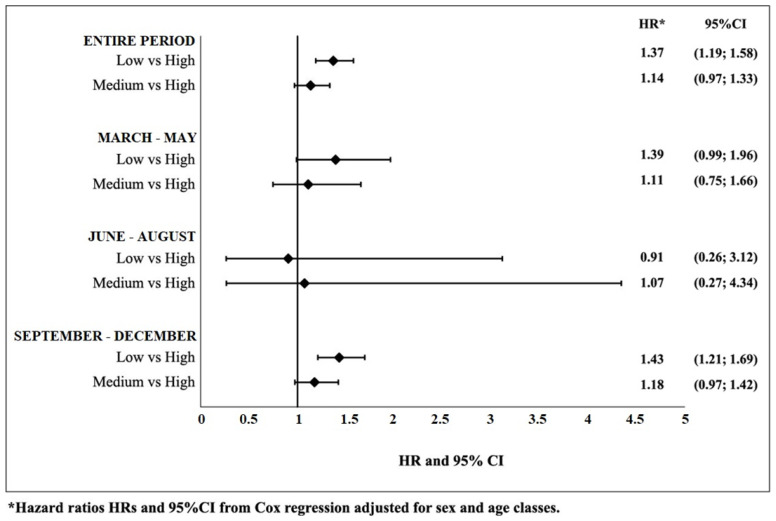
Association between educational level and mortality within 30 days of the onset of SARS-CoV-2 infection for the entire period and in the three phases of the pandemic.

**Table 1 jcm-11-00877-t001:** SARS-CoV-2 infections by educational level, sex, and age. Crude cumulative incidence (CCI) with 95% CI. Rome, 35+ years old, 1 March to 31 December 2020.

	Participants		SARS-CoV-2 Infection		CCI ‰	95% CI	
	N = 1,538,231	100%	N = 47,736	100%	31.00	30.80	31.30
EDUCATIONAL LEVEL	630,745	41.00	19,999	41.90			
Low					31.71	31.27	32.15
Medium	555,885	36.14	17,707	37.09	31.85	31.39	32.33
High	351,601	22.86	10,030	21.01	28.53	27.97	29.09
SEX	684,453	44.5	22,384	46.89			
Male					32.70	32.28	33.13
Female	853,778	55.5	25,352	53.11	29.69	29.33	30.06
AGE CLASSES	243,054	15.8	8487	17.78			
35–44					34.92	34.18	35.67
45–54	377,845	24.56	13,653	28.60	36.13	35.53	36.74
55–64	340,400	22.13	11,053	23.15	32.47	31.87	33.08
65–74	267,232	17.37	6276	13.15	23.48	22.91	24.07
75–79	115,456	7.51	2653	5.56	22.98	22.12	23.87
80+	194,244	12.63	5614	11.76	28.90	28.16	29.67

**Table 2 jcm-11-00877-t002:** SARS-CoV-2 infections and deaths within 30 days from infection by educational level, sex, and age. Crude Mortality Rates (CMR) with 95%CI. Rome, 35+ years old, 1 March to 31 December 2020.

	SARS-CoV-2 Infection		Death within 30 Days		CMR ‰	95% CI	
	N = 47,736	100%	N = 2281	100%	47.78	45.86	49.79
EDUCATIONAL LEVEL	19,999	41.90	1614	70.76			
Low					80.70	76.86	84.74
Medium	17,707	37.09	438	19.2	24.74	22.53	27.17
High	10,030	21.01	229	10.04	22.83	20.06	25.99
SEX	22,384	46.89	1284	56.29			
Male					57.36	54.31	60.59
Female	25,352	53.11	997	43.71	39.32	36.96	41.84
AGE CLASSES	8487	17.78	10	0.44			
35–44					1.18	0.63	2.19
45–54	13,653	28.60	48	2.1	3.52	2.65	4.67
55–64	11,053	23.15	142	6.23	12.85	10.90	15.14
65–74	6276	13.15	411	18.02	65.49	59.45	72.14
75–79	2653	5.56	345	15.12	130.04	117.02	144.51
80+	5614	11.76	1325	58.09	236.03	223.64	249.08

## Data Availability

Health Information System and Integrated Surveillance System of SARS-CoV-2 infections contain confidential data. We obtained access to these data from the Lazio Region within the framework of the research project DETECOVID (COVID-2020-12371675) and worked on Lazio Region servers. The full dataset can be accessed from the regional servers only, and cannot be published for legal reasons.

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
