# Peer review of "Incidence of SARS-CoV-2 Infection and Related Mortality by Education Level during Three Phases of the 2020 Pandemic: A Population-Based Cohort Study in Rome"

_jcm, 2022, doi:10.3390/jcm11030877_

Round 1

Reviewer 1 Report

Dear authors,

The article is well written, but the language needs to be modified at places to avoid plagiarism since it is same as the reference articles.

Further though there is no new message the script conveys; it does re-emphasize the inequality in available health care based on level of education and socio-economic status.

Reviewer 2 Report

Thank you for the interesting manuscript considering impact of socioeconomic variables on the outcomes of COVID-19 infection. The manuscript makes potentially an important contribution to the literature, but in my view would benefit from better organised introduction and methods sections with clearer explanations of study background, aims, and methods. Number of issues that were unclear while reading the text become only clearer in the discussion as a lot of relevant literature was only first introduced in there. Please see more detailed comments below.

Abstract

Conclusions section should be improved to be more specific and information added about what new this research is adding to knowledge about influence of socioeconomic factors influencing the risk of illness (in this case COVID-19).

Introduction

Line 66: “on the other side…” the sentence is long and difficult to understand.

Line 68: It comes as a surprise that this is the second possible pathway – as it wasn’t clear in the text before that pathways to the expected effects are discussed.

Line 78: Please add reference.

Line 80: Is there something missing from the sentence? Not sure what the authors want to say in here.

Paragraph from line 80: Overall, this paragraph is hard to understand. It is difficult to understand whether authors speak about the COVID-19 as a disease or pandemic as such. It should be made clear whether the information relates to influence of socioeconomic determinants during the course of an individual’s illness or pandemic. In addition, the theoretical background should be expanded to explain the context of the research and how this adds to knowledge. Also, based on the extensive background material regarding illness and socioeconomic determinants of health, it was surprising to find that the authors had not formulated any hypothesis to test.

Methods

Methods is very limited with the information and it is difficult to understand how the research was conducted. Authors should include considerably more details about how the research was conducted. T is stated that the research was prospective, but there are indications that the actual design was retrospective. There is no clear definition how the different pandemic-phases are defined. Also, it would be helpful to explain why the three consecutive period and not the whole 9 months period?

Why the cut-off age of 35? Is this to do with the census information? Please could it also be specified whether this was the age at the time of census or at 2020. It is very unclear how the information in the census was used – level of education is not the only variable linked to social status. Did the authors considering using additional variables such as level of income o occupation? Considering that access to education and employer requirements are likely to have been changed overtime (I.e. younger generation better educational access – increased employer demand for degrees etc.) – did the authors consider this in their analyses? In addition, the text should be clearer about why incident of the covid-19 is an outcome measure – especially as authors point out that only those tested and residing in Rome in 2011 were included. Also, there is repetition on outcomes and follow-up.

Statistical methods

Considering multiple testing of same variables – was the p-value adjusted?

Results

Large amount of repetition of result both in text and tables should be avoided.

Discussion

Number of references has been included in the discussion that should have already been included in the introduction. This would have helped the reader to understand the context, content, and methods of the study better – before reaching discussion. Also, the beginning of the discussion resembles results section – no numeric results should be included in the discussion.

Line 74: Reference

Line 121: These conclusions are not based on the available data in this study.

Reviewer 3 Report

The manuscript is well written, clear and well organized. 

In case you can provide data on households size (e.g. number of children) of the different groups it would add another perspective. 

Also, I suggest to take into account the "clustering effect" of infectious diseases. Can you use contact-tracing data to correct for the clustering? 

Reviewer 4 Report

This is a cross-sectional analysis of a large-scale cohort data using the anonymized health information data, COVID-19 data, and the census data that are already linked by personal identifier. The system should be introduced earlier in the manuscript to explain your specific standpoint as personnel in the regional health service and to explain the superiority of such a data system in Italy. As stated in the Strength and Limitation, the data is not usually available on such a large scale without intervening the personal information.

It is appreciated that you used the multivariate log-binomial and Cox regression models to validate the statistical significance in a cross-sectional study. However, the results could be easier to understand if you indicate the significant differences using the p values in the tables. Aging is also a significant determinant of the outcome as shown in Fig. 1B. I appreciate it if you could confirm that the distribution of educational levels among the age group did not have any statistical difference. It might be possible that the high age group might be associated with lower educational attainment.

You should describe that the difference was not statistically significant between the middle-level educational group and high-level educational group in any phase suggesting that the upper secondary school level may be sufficient to develop the health literacy and economical circumstances to cope with COVID-19. In the interrelationships of the sustainable development goals, poverty is a deprivation of opportunities for education, business, and many other social determinants of health. Hence, the low-level group and the high-level group have much more differences other than the levels of education as you described. At the same time, it is possible that older people have a similar vulnerability with people with low-level education. I appreciate your careful discussion.

I am not sure about the necessity of Table 2. The description in the manuscript also does not add any new information over the findings in Fig. 1A and 1B. Why do you show the gender difference and not the age difference in the table of age- and gender-adjusted CCI and DMR?

Minor points

  • Line 102: It is better to bring the statement of IRB review Section to indicate the specific position of the authors and the Health Information system that already made it possible to assess the educational certification without touching any personal data of the citizen. Because at this moment, it is not clear how anonymized identifiers can allow record-linkage. Add more explanation on the method of anonymization and identification of how you can link the educational level of a person with or without COVID-19 infection.
  • Line 114: Add an explanation on what you mean by "follow up". In Line 92, you describe that "all residents were followed through the Regional Health Information System" suggesting that an already unique identifier was assigned to every citizen and the system is actively/passively gets the health information of that person periodically. Is it correct?
  • Line 140: You should use "mortality rate" instead of "mortality risk" because this is a proportion of people who died of COVID-19 within the patients (people at risk). If you insist to use "risk", indicate the reason why.
  • Line 150: Do not start a sentence with a number.
  • LIne 152: Indicate whether the difference was statistically significant or not, rather than showing the same values of CIs that are already shown in the Table.
  • Table 1: It seems "crude mortality rate (CMR)" is correct usage of terminology. Correct the Title of the table and "CRM" in the column.
  • Table 1: Indicate the statistical difference between the categories with p values using univariate analysis. Indicate where the difference is statistically significant.
  • Page 9, Line 3: It is not clear what "contrasting results" mean. It seems to me that Fig.1A is simply showing that infection occurred mainly in the Sep-Dec phase. Fig. 1B is showing the crude mortality rate within the test-positive patients suggesting that March/May, and June/Auggust phases had very higher mortality rate because of the unknown characteristics of COVID-19 and difficulty of treatment. Describe the each determinant of outcome, after that.
  • 2 and 3: Correct to "December"
  • Page 14, Line 38: Do you mean “(RR=0.71 and 0.64)” at the “first” phase as shown in Fig. 2? I appreciate it if you could indicate the basis of the notion as (Fig. 2) etc.

Round 2

Reviewer 2 Report

Many thanks for the authors for their diligent attention and consideration of the comments.

After reading the revision, I would have just few minor points for consideration.

Line 83: ...awareness and habits of residents. Not sure what the authors refer to in here? The research did not really look athe residents habits as such?

Use of word "subject" - this is generally discouraged - participant/patient/individual...

Line 109: please could you adda reference.

In the Tables (e.g. 1 & 2 ) Please consider - usuall described age groups - not classes.

Reviewer 4 Report

Thank you for the revision.

Author Response

We thank Reviewer 4 for his/her comments.